# Quality by Design-Based Methodology for Development of Titanate Nanotubes Specified for Pharmaceutical Applications Based on Risk Assessment and Artificial Neural Network Modeling

**DOI:** 10.3390/pharmaceutics17010047

**Published:** 2025-01-01

**Authors:** Ranim Saker, Géza Regdon, Krisztina Ludasi, Tamás Sovány

**Affiliations:** Institute of Pharmaceutical Technology and Regulatory Affairs, University of Szeged, Eötvös u 6, H-6720 Szeged, Hungary; rnmsaker@gmail.com (R.S.); geza.regdon@pharm.u-szeged.hu (G.R.J.); krisztina.ludasi@szte.hu (K.L.)

**Keywords:** titanate nanotubes, hydrothermal treatment, quality by design, ANN, risk assessment, pharmaceutical applications

## Abstract

Background: Nanotechnology has been the main area of focus for research in different disciplines, such as medicine, engineering, and applied sciences. Therefore, enormous efforts have been made to insert the use of nanoparticles into the daily routines of different platforms due to their impressive performance and the huge potential they could offer. Among numerous types of nanomaterials, titanate nanotubes have been widely recognised as some of the most promising nanocarriers due to their outstanding profile and brilliant design. Their implementation in pharmaceutical applications is of huge interest nowadays as it could be of fundamental importance in the development of the pharmaceutical industry and therapeutic systems. Methods: In the present work, a risk assessment-based procedure was developed and completed using ANN-based modeling to enable the design and fabrication of titanate nanotube-based drug delivery systems with desired properties, based on the critical analysis and evaluation of data collected from published articles regarding titanate nanotube preparation using the hydrothermal treatment method. Results: This analysis is presented as an integrated pathway for titanate nanotube preparation and utilization in a proper way that meets the strict requirements of pharmaceutical systems (quality, safety, and efficacy). Furthermore, a reasonable estimation of the factors affecting titanate nanotube preparation and transformation from traditional uses to novel pharmaceutical ones was established with the aid of a quality by design approach and risk assessment tools, mainly an Ishikawa diagram, a risk estimation matrix, and Pareto analysis. Conclusions: To the best of our knowledge, this is the first article using the QbD approach to suggest a systematic method for the purpose of upgrading TNT use to the pharmaceutical domain.

## 1. Introduction

Titanate nanotubes (TNTs) have attracted considerable attention during the past three decades since they were first presented in 1996 [1]. Numerous research works have been published focusing on the usage of titanium dioxide (TiO_2_)-derived substances, including TNTs, in several applications, such as chemical catalysis, water purification, energy saving, and solar cells, but a very limited number of works have focused on their potential use in the pharmaceutical field as drug carriers (Figure 1).

Their unique tubular morphology and higher surface area compared to their spherical precursors, in addition to the existence of hydroxyl groups on their vast surface, provide them with preferred features for medical/pharmaceutical applications. Furthermore, their surface properties may be tailored by functionalization or by grafting of specific molecules for diagnostic or therapeutic purposes [2,3]. Their tubular morphology also promotes cell internalization compared to their spherical counterparts [4]. Furthermore, their safety profile introduces them as an attractive candidate for drug delivery by loading them with therapeutic agents inside their tubular cavity or on their surface [5,6]. TNTs also present good processability in terms of flowability, compactibility, and compressibility, which has a huge, positive impact regarding manufacturing at the industrial level [7,8]. With these outstanding features, TNTs could be the key solution to overcoming the fundamental challenges of nanomedicines faced during pharmaceutical development, such as poor tabletting, solubility, and bioavailability. Therefore, they deserve more attention and a precise investigation so they can be introduced into therapeutic practice as efficient tools. However, this task is becoming more and more challenging due to the tremendous development happening around us in the modern world, which raises the bar, thus posing an urgent need to apply the innovative tools discovered recently (statistics software, computational systems, machine learning (ML), artificial intelligence (AI), etc.) to serve the evolution of the pharmaceutical domain as traditional ways are no longer sufficient to achieve this target. These intelligent tools should certainly consider the most important standards in the pharmaceutical industry: quality, safety, and efficacy.

Continuously, pharmaceutical authorities emphasize building these criteria into products by design rather than just testing them eventually. This concept was the essence of what is called today the quality by design (QbD) approach, which has been gaining more and more attention due to its fundamental contribution to pharmaceutical research. QbD tools revolutionized the formulation field with their potential to draw guidelines to obtain the requested specifications and design a workspace within which the quality of the final product could be guaranteed [9,10]. The application of this novel approach is widespread and includes different types of drug delivery systems (polymeric microspheres, nanocrystals, pellets, nanostructured lipid carriers [11,12,13,14]) and several routes of administration (oral, dermal, parenteral, intranasal [13,14,15,16]).

This accelerated development in pharmaceutical research has not stopped with only the application of the QbD approach but is also supported by the application of artificial neural networks (ANNs), one of the most popular types of ML tools, as they can overcome the numerous limitations facing conventional methods, in addition to saving time, effort, and money, still with impressive outcomes. ANNs have been applied successfully in solid dosage forms for several purposes, such as modeling of tablet coatings [17], prediction of dissolution profiles and hardness [18], optimization of the spray-drying process [19], modeling of powder flow [20], and optimization of controlled/sustained release tablets [21,22]. Their application has also been extended to the formulation of other dosage forms/carriers (liposomes, hydrogels, emulsion, etc.) [23,24,25] and for multiple delivery routes (oral, transdermal, etc.) [21,22,26]. Based on the above discussion, the combination of the revolutionary concepts of QbD and AI—such as ANN—could write a new chapter in the development of pharmaceutical sciences and lead to a new era of research and industry [18,27].

For this reason, the main aim of this research is to propose a systematic procedure to shift the use of TNTs from their traditional applications to the pharmaceutical field using the tools of QbD and ANNs. In this context, this paper will suggest a multi-step procedure to achieve this goal and discuss in detail the appropriate parameters that could be implemented/adjusted to suit the special criteria of the therapeutic domain. This suggested methodology uses risk assessment tools based on the previous experience of the research group and the available data in the literature. The importance of the research team’s previous experience with the evaluated topic should be emphasized as it will have a major impact on the selections/decisions made and the reliability of the evaluation produced. On the other hand, critical evaluation of previously reported information is also crucially important as it will help to build a solid background that enables a precise assessment of the risks associated with the utilization of TNTs as drug carriers in order to create applicable guidelines for the development of TNT-based drug delivery systems. It is worth mentioning that risk assessment is not a static analysis but can be modified and updated regularly based on upcoming research outcomes and continuously published articles [28,29,30].

In this paper, the hydrothermal treatment method, a chemical reaction that involves material crystallization from a highly concentrated alkaline solution at elevated temperature for a specific duration, was chosen for the preparation of TNTs due to its numerous advantages over other techniques (sol–gel process, electrochemical anodization, template-assisted method, vapor-liquid-solid growth) as it is simple, cost effective, environmentally friendly, scalable, and controllable [31,32,33,34,35]. In addition, the authors have experience with this method from their previously published research [7,8,36,37,38].

By using this method, the general characteristics of the obtained TNTs seem to be ideal for pharmaceutical applications. Furthermore, their characteristics (e.g., diameter, length, surface area, etc.) can be manipulated by varying the production conditions; thus, with an optimized process, TNTs may be obtained with specific properties preplanned for targeted application [32]. Regarding TNTs, their tubular morphology, nano-size, and crystal structure, in addition to their surface characteristics (area, chemistry, coating) and preparation yield would highly impact their use as potential drug carriers by affecting the selection of active pharmaceutical ingredient (API), dosage form, and route of administration, as well as their stability, safety profile, and drug release pattern.

To the best of our knowledge, no systematic approach was previously proposed to draw basic lines for the use of TNTs as nanocarriers for therapeutic agents starting from the point of hydrothermal preparation and ending with a final dosage form that can be presented to the market/targeted patients. To achieve this goal, the QbD approach and risk assessment tools in combination with ANNs have been adopted for the first time to push the use of TNTs toward the pharmaceutical field, which is an innovative, under-explored application.

## 2. Materials and Methods

### 2.1. Risk Assessment

#### 2.1.1. Definitions in QbD Methodology

The QbD approach is a multi-step procedure that starts from defining the quality target product profile (QTPP), which is a general abstraction of indicators of quality, safety, and efficiency of a product.

In the second step, the general QTPP indicators should be translated to measurable or calculable physical, chemical, biological, or microbiological parameters, called critical quality attributes (CQAs). Furthermore, the target values and acceptable tolerance ranges of CQAs should be determined and considered as the product design space (DS).

The third step is the identification of the material and process-related factors (critical material attributes (CMAs) and critical process parameters (CPPs), respectively) that may affect CQAs.

Finally, the interdependence between CPPs, CMAs, and CQAs should be determined, and the severity, probability, and detectability of the impacts caused by the changes in CPPs or CMAs should be estimated.

#### 2.1.2. Methods of Literature Survey

CQAs, CMAs, and CPPs were selected using previous experience and literature data. Relevant scientific databases: Web of Science, Scopus, PubMed, etc., were searched using the keywords titanate nanotubes, synthesis, hydrothermal treatment, drug delivery, toxicity, etc. A database including the investigated CMAs, CPPs, and CQAs was created after critical evaluation of the gathered data, and this database was used for supporting the risk assessment and served as basis of the ANN modeling.

### 2.2. Quality Tools

Various quality tools were applied during the risk assessment. An Ishikawa diagram was used for the identification of CPPs and CMAs, a risk estimation matrix was used to determine their interdependence, while Pareto analysis was used to quantify the risk regarding various CQAs.

#### 2.2.1. Ishikawa Diagram

The Ishikawa or fishbone diagram can be used to determine the root causes of a problem or the factors affecting a specific event. In this research, the factors affecting different steps and profiles (TNT preparation, API incorporation, functionalization, and safety profile) were identified. These factors should be seriously taken into consideration during the development process to achieve the desired results. This cause–effect diagram is very useful for preplanning the upcoming experimental work and highlighting the most prominent factors by visualization when a limited amount of quantitative data is available.

#### 2.2.2. Risk Estimation Matrix and Pareto Analysis

The risk estimation matrix represents the correlations between QTPPs and CQAs or between CPPs/CMAs and CQAs. The interdependence rating, which describes the relationship between the parameters, was determined using a three-level scale, high (H), medium (M), or low (L) dependence, and they appeared in the interdependence tables using the colors red, yellow, and green, respectively.

LeanQbD software (Version 1.3.6., 2014 QbD Works LLC, Fremont, CA, USA) was used for the evaluation. In the software, the qualitative three-level scale used for the estimation is linked to a selectable numeric scale which, at the end of estimation, gives the severity score of the evaluated risk factors based on mathematical calculations. After the categorization of the interdependence, the risk occurrence rating of the CPPs was made, applying the same three-grade scale (H/M/L) for the analysis. As the output of the initial RA evaluation, Pareto diagrams were generated by the software, presenting the numeric data and the rankings of CQAs and CPPs/CMAs according to their potential impact on the aimed final product. Pareto analysis is a statistical tool that enables effective data evaluation and facilitates a more reliable decision-making process by determining and prioritizing the factors having the greatest effect on the studied system. This method has crucial importance in the screening of numerous factors affecting the quality of the final desired product, and attention should be directed first toward the factors with the highest influence.

### 2.3. ANN Modeling

ANN modeling was performed using Statistica v.14.0.1.25 (Tibco Software Inc., Palo Alto, CA, USA). The full dataset collected from the literature is displayed in Appendix A. It is easily visible that the range of the discussed data is very versatile among the various papers. After careful data curation, two outputs were identified in which the existing data enable suitable modeling. The morphology and the specific surface area of the obtained products were then handled as two different datasets for classification and regression-type modeling, and the corresponding datasets are displayed in Appendix A, respectively. The range of input variables was the same for both models. After setting the datasets, the minimum and maximum values of each input and the output parameters were identified manually, and these cases were included into the training subset to avoid the need for extrapolation during testing or validation of the model. The remaining data were then randomly distributed into the training, testing, and validation subsets, where the sizes of the training, testing, and validation subsets were 70%, 15%, and 15% of the full dataset, respectively.

Feed-forward, back-propagation multilayer perceptron networks were developed in both cases. The networks were trained with the BFGS algorithm. The possible range of the number of hidden neurons was set according to the following equation (Equation (1)):1 ≤ *n* ≤ *I* + *O* + 1(1)
where *I* is the number of inputs, *O* is the number of outputs, and *n* is the number of hidden neurons; thus, the hidden neuron number varied from 1–20 for the classification and 1–12 for the regression network, respectively.

A multistart method including 10,000 networks was applied using the automated neural networks module of Statistica, including a training approach to screen the best performing network with different initialization patterns and activation functions for hidden and output neurons. The training was stopped when the root mean square error (RMSE) of the test dataset reached its minimum. The 10 best performing networks from each multistart run were retained for further analysis.

The prediction performance of the networks was evaluated based on network perfection, which is the mean R^2^ of the observed vs. predicted data of each output neuron, and on the RMSE of predictions on the validation subset.

## 3. Results and Discussion

According to literature data, hydrothermally synthesized TNTs may be appropriate candidates for delivering APIs and for providing an efficient tool for use in healthcare systems thanks to their good biocompatibility and small size, which is hard to detect by the immune system [39]. However, the identification of QTPPs and related CQAs (Table 1 and Table 2, respectively) is an essential step in the development procedure of a product as they are application-related parameters that are always different and unique for every case.

The properties of manufactured nanotubes are dependent on the properties of the starting materials and on the operating conditions during the hydrothermal reaction. Furthermore, they are also dependent on the conducted post-treatment steps.

In the present investigation, QTPPs include appropriate physical properties and stability; therapeutic effects; and appropriate pharmacokinetics, including absorption followed by drug release and elimination, ensuring safe use for patients. Accordingly, morphology, which is unique in the case of nanotubes, size, type, crystal structure, specific surface area (SSA), surface characteristics, yield of preparation, and drug loading were chosen as CQAs.

To obtain TNTs with the required properties for a specific application, such as a pharmaceutical one, the hydrothermal synthesis method should be carefully optimized. As previously mentioned, in this chemical reaction, the starting precursor is subjected to high temperature in a concentrated alkaline medium for a specific duration, so the multiple parameters that are deeply involved in this procedure are highly responsible for the result.

For example, temperature was the most discussed factor affecting the success of the preparation process as it serves as an energy supply to help the intermediate nanosheets to curl up and form nanotubes. Therefore, it should be used in the appropriate range, as too low a temperature would not be enough to transform the starting nanoparticles/intermediate nanosheets to nanotubes. On the other hand, too high a temperature would also pose a problem, with several morphologies obtained other than the tubular one. This information could be of great importance if other morphologies are the morphologies of interest. In this case, the temperature can be raised up to create nanofibers (>150 °C) [40], nanorods or nanoribbons (>160 °C) [41,42,43], nanofibers or nanobelts (>170 °C) [44], and nanorods (>180 °C) [45]. According to the literature, the temperature range between 130 and 150 °C is the most repeated one in several research works, which represents a strong point of agreement and confirms the formation of complete nanotubes within this range. In addition, increasing the temperature within this range would increase the diameter, SSA, production yield, and enhance crystallinity without negatively affecting the requested tubular structure [40,42,45,46]. Together, these specifications could be an advantage to be invested for the benefit of drug delivery systems.

Besides high temperature, TNT preparation involves the usage of a highly concentrated alkaline medium, which supports the thermal energy to provide the required force for rolling up the sheet phase into a tubular one. Based on this, the concentration of this medium should also be within an appropriate range (10–15 M), as too low a concentration would result in unreacted powder/untransformed sheets, while too high a concentration would result in low yield [42,47,48]. It is worth mentioning that this treatment should take sufficient time to give the intermediate phase an opportunity to transform into nanotubes and then to give the obtained nanotubes enough time to grow and elongate their length. Fifteen hours is probably the optimal time for the formation of pure nanotubes, with the possibility of increasing their length by increasing the treatment time to 24 h. However, caution should be taken as longer duration could lead to the formation of other morphologies, such as nanoribbons [49,50]. Furthermore, after a specific point, no length increase occurs, so further treatment will be just a waste of energy, which would subsequently have a negative impact on the economic cost [51].

The pressure inside the sealed autoclave during the reaction is one of the variable preparation conditions, but its impact on nanotube formation was not sufficiently discussed [52,53]. Morgan et al. suggested that the pressure effect could be excluded from the significant factors affecting the formation of TNTs [43].

After the hydrothermal reaction is finished, the post-treatment processes (acid washing, calcination) take place and play a major role in preserving the desired morphology and obtaining the targeted characteristics. According to several publications, washing with acid could affect the size of TNTs and enhance their crystallinity [54]. However, two critical factors were deeply discussed in literature regarding this process. Firstly, the pH of the washing solution was suggested to be kept between 2 and 4 to obtain a high yield and high SSA [55]. The second one is the concentration of the used acid, on which there is disagreement, as few studies recommended the use of low concentrations because high ones would destroy the tube-like morphology (>0.01 M) [56] and result in the formation of granules (>0.2 M) [57] or clumps (>2 M) [52,58], while others disagreed and indicated that high concentrations (0.5–1 M) were the optimal range to use [52].

Another post-treatment process (calcination) can be used to enhance the crystallinity of the resulted materials, but caution should be taken not to negatively affect the morphology and thus the SSA. The safest range to work in is between 200 and 400 °C, as increasing temperature within this range would enhance crystallinity without negatively affecting the tubular structure [42,49,59,60,61]. Higher temperature could lead to destroying the tubular structure (>450 °C) [61] and complete collapse of the nanotubes into irregular shapes (>500 °C) [60], nanorods (>540 °C) [42], nanoparticles (>600 °C) [49,59], or aggregates (>800 °C) [51,62].

It was noticed that the discussed results in the literature were not always comparable as different operating conditions would lead to different and conflicted results. These differences make it difficult to establish strict guidelines for TNT preparation with prespecified characteristics. This means that studying the effect of different preparation factors separately poses a research challenge because they act as a full combination rather than independent ones. For example, a lower alkaline medium concentration could be compensated by the utilization of higher temperature [43]. In the same context, lower temperature could be applied as long as prolonged treatment is used [50]. Moreover, numerous factors during the preparation reaction could shorten the required duration for TNT formation, such as stirring, smaller particle size of the starting material, or using less stable, higher energy anatase phase precursors [47,61,63]. For this reason, the required parameters should be modified together to achieve the appropriate balance. This could be performed according to the requested final specifications of the prepared TNTs and according to the available laboratory instruments/equipment and the ranges within which they can work.

Careful evaluation and curation of the existing data enables us to better understand and model how to manipulate the operating conditions to achieve a specific purpose or affect a specific property of the resulting TNTs. For instance, increasing the factors (temperature, alkaline concentration, acid concentration) within the previously specified limits and decreasing the particle size of the starting precursor would increase the SSA, but the opposite would happen by increasing the treatment time [42,46,47,64,65,66]. This conclusion is very important because high surface area is one of the most significant characteristics of TNTs due to its direct impact on their possible functionalization and drug loading, so it could serve as an advantage for pharmaceutical applications as it successfully served earlier in energy saving and chemical catalysis.

Starting from a crystalline precursor (anatase) that promotes the thermal stability of the resulting nanotubes [67], washing them with acid will reduce the sodium content and resistance to temperature, resulting in H-TNTs that are thermally less stable than Na-TNTs [55,58,65,66,68].

The concentration of the used precursor also has an impact on the specifications of TNTs. High concentration increases the length/aspect ratio and diameter of the prepared TNTs but negatively affects the SSA [40,69]. It could affect morphology; therefore, manipulating the Ti/Na molar ratio is an important step in shaping the structure of the resulting material into the desired direction (nanotubes, nanowires, nanofibers) [49].

After titanate nanotubes have been prepared properly according to the planned design and predetermined specifications, modifying their vast surface is a well-known technique for giving them secondary characteristics that differ from their original ones. Surface modification could be the key solution to overcoming several problems facing the upgrading of TNT use to the pharmaceutical field. For example, surface functionalization using hydrophobic moieties could be a suitable way for enhancing the poor permeability of TNTs, which originates from their hydrophilic properties and hydroxylated surface [38]. PEGylation could also present several advantages as it could be an appropriate solution for the aggregation problem in addition to making them undetectable by the immune system, prolonging their circulation time as well as reducing their toxicity [2,70,71]. Determining the purpose of surface functionalization is an unavoidable step, as it would affect the type of molecule that is going to be fixed on the surface of TNTs and the type of interaction that will be created. This interaction is highly dependent on the chemical structure of the functionalizing agent and the chemical bond that is going to be created, which is preferred to be a strong, long-lasting bond like covalent or ionic bonds rather than hydrogen ones [38,72]. Hydrogen bonds could be more favorable in some cases, such as during API incorporation, as weak interaction between TNTs and APIs would not strongly affect drug release unless the modification of drug release is the main target of incorporation [36].

Drug incorporation is also considered to be a challenging step during the development of nanocarrier drug delivery systems as multiple factors could affect the success of this procedure. Based on the authors’ previous experience, the chemical properties of the used drug and solvents are essential during TNT-drug composite formation. Solvent polarity (protic/aprotic nature) and volatility would highly affect the success of composite formation as they would affect the possibility of drug solubilization, the strength of solvent–drug interactions, and the ability to remove solvent from the system. In the ideal state, the requested drug–carrier interaction should be stronger than the one created between the drug molecules themselves or between the drug and the solvent. However, the strength/type of TNT–drug interaction should be further investigated as it could be determined according to the desired release type (immediate or sustained) [36,37].

Finally, presenting titanate nanotubes as possible candidates for drug delivery systems is strongly dependent on their safety profile, which is still a matter of debate. However, several points have been thoroughly discussed in the literature and could be considered as a starting point to build safer nanocarriers for therapeutic use. Morphology is one of these critical points affecting toxicity. It is true that the high surface area related to the tubular morphology is considered an advantage, but it also promotes cell penetration and the subsequent toxic effects [73]. Surface chemistry/coating could also play a significant role in determining toxicity as changing the existing Na^+^ ions on the surface of TNTs to H^+^ or Mg^2+^ will increase their toxic effects [38,74], while applying a specific type of hydrophilic coating, such as PEGylation, could be a proper way to reduce aggregation and toxicity [70,72]. The applied concentration is also crucial as increasing the used dose of these nanotubes would increase the impact of their hazard [73]. In addition, the crystal structure could also affect the safety profile of TNTs as TiO_2_ can exist as three different structures (anatase, rutile, and brookite), of which anatase appears to exert the highest toxic effects [75,76,77,78]. Understanding these elements and their direct impact on toxicity would help researchers to optimize the preparation procedure and the subsequent steps to obtain a safer final product.

The CMAs and CPPs identified to affect the development of TNTs as possible drug carriers for pharmaceutical applications are discussed in Table 2 and an Ishikawa diagram is proposed (Figure 2) to summarize the necessary materials and processes for the utilization of TNTs in drug carrier systems starting from the moment of preparation until adjusting the surface properties by functionalization, loading with therapeutic molecules, and finally assessing the safety profile in order to incorporate these synthesized nanomaterials into final dosage form (oral, dermal, etc.).

Most of the selected CQAs are attributed to multiple elements of QTPPs. Therefore, a risk estimation matrix (REM) was created (Table 3) with the aid of LeanQbD v1.3.6. software to estimate the interdependence ratings between the collected elements. Furthermore, Figure 3 visualizes the priority of CQAs of hydrothermally synthesized TNTs as drug carriers depending on the chosen QTPPs. According to Pareto analysis, the most significant CQAs were surface characteristics, morphology, SSA, size, type of TNT, crystal structure, drug loading, and yield.

The results of REM and Pareto analysis correspond well with the available data in the literature, which repeatedly emphasized the importance of the tubular morphology and large surface area of TNTs in their possible usage as drug carriers due to the unique characteristics that come along with these two parameters, such as high cell internalization, as well as the ability to load APIs inside the tubular cavity or on their vast surface.

However, according to Pareto analysis, the properties of this vast surface, which can be controlled through modification with a proper molecule, is the most important as it can determine the success of the whole transformation procedure by its fundamental impact on the chosen QTPPs. For example, surface modification could be used to control/enhance the release rate of loaded drugs [79,80,81] and to enhance the pharmacokinetic properties (for example, absorption), which would have a huge influence on determining the used dosage form/route of administration and a high positive impact on the therapeutic effect [38]. It could also be used to improve the safety profile and decrease toxicity [81].

The same evaluation was performed again by creating a risk estimation matrix (Table 4), as well as conducting Pareto analysis (Figure 4), to study the importance and ranking of CPPs and CMAs during the transformation of hydrothermally synthesized TNTs into possible drug carriers. The most important CPPs were the pH of the washing solution, reaction temperature, reaction time, calcination temperature, and stirring, while the important CMAs were precursor concentration, functionalizing agent, alkaline medium concentration, precursor particle size, precursor crystal structure, drug type, and used solvent.

The results of this evaluation agreed well with the literature data as most of the performed studies reported that the washing step is an unavoidable phase in the preparation of TNTs [54,62,82], along with selection of the appropriate set of temperature [40,41,42,43,44,45,46], time [49,50,51], and precursor concentration [40,49,69].

The pH of the washing solution is a significant factor in the washing process, which serves as the complementary step to obtain the desired tubular morphology of TNTs if the reaction conditions are not sufficient to achieve this. It would also have a major impact on most selected CQAs, such as the size of the resulting nanotubes, their type, SSA, and the preparation yield. For these reasons, it is justified why the pH of the washing solution is in the top ranking of CPPs, followed by reaction temperature and time, which also have a high impact on the majority of CQAs, like morphology, size, SSA, and yield. Moreover, these previously mentioned CQAs were influenced in the same way by the top-ranked CMA, which was precursor concentration. Again, these findings correlated well with the data presented in the literature, as the resulting top-ranked factors were the most discussed factors in the literature, which supports their priority classification according to the results of Pareto analysis.

As discussed above, a dataset (Appendix A) was created based on the existing literature [4,5,41,47,48,50,51,52,53,55,57,58,59,60,61,62,63,64,65,67,68,69,73,74,79,80,81,82,83,84,85,86,87,88,89,90,91,92,93,94,95,96,97,98,99,100,101,102,103,104,105,106,107,108,109,110,111,112,113,114,115,116,117,118,119,120,121,122,123,124,125,126,127,128,129,130,131,132,133,134] to perform ANN modeling on possible CQAs. The most discussed CQAs in the literature were the morphology and SSA of the obtained product, which enabled the gathering of data sufficient to build an ANN-based model for the prediction of the possible outcome of a synthesis and post-treatment process. Furthermore, the global sensitivity analysis also enabled us the check the validity of the results obtained during the risk assessment procedure.

The structure (e.g., the percentages of rutile, anatase, and amorphous phases), surface area, and particle size of the starting precursor were considered as input CMAs, while the temperature and NaOH concentration of the reaction medium, the reaction time, the acid concentration of the washing liquid, and calcination temperature were selected as input CPPs. Most researchers used HCl in the washing step, but in some cases HNO_3_ was applied. This meant a considerable difficulty in that the morphology of the obtained product was described in the literature with high versatility, which necessitated intensive data curation regarding this aspect. Besides the nanotubular morphology (1), the terms nanorods (2), nanofibers (3), and nanowires (4) were given to those products where one direction of particle growth was featured, such that products with structures with an increasing length to diameter ratio were named nanorods, nanofibers, and nanowires, respectively. The classification of those particles where growth was featured in two directions were unified under the term nanoribbons (5), while spherical products were classified as nanoparticles (6) or spherical agglomerates (7). Two additional classes were made for those cases where no conversion (8) of the starting material was observed, or the product obtained was a mixture (9) of particles with different morphologies.

The best performing network for the classification of morphology had 10 input, 10 hidden, and 9 output neurons, where the activation of the hidden and output neurons were based on identity and softmax functions, respectively. The network structure (e.g., the weights between various neurons) is displayed in Appendix A. The perfection of the classification for the training, testing, and validation subsets was 85.71%, 61.9%, and 76.19%, respectively. The results of the classification on the validation dataset are displayed in Figure 5. Most of the misclassified cases predicted the formation of nanotubes, which may have been due to over-representation of this class in the training subset, and a considerable increase in classification accuracy may be expected with a more balanced dataset. The results of the other retained networks showed high consistency with these results.

The results of the global sensitivity analysis partially supported the results of the risk assessment, but the highest impact was exhibited by the calcination temperature, which was followed by particle size and the structure of the precursor material (mostly by the amorphous content), with approximately equal contributions, while the NaOH content and the temperature of the reaction medium took third place, again with approximately equal contributions.

In the case of modeling of SSA, the best performing network had 10 input, 10 hidden, and 1 output neurons, where the activation of the hidden and output neurons was based on tanh and logistic functions, respectively. The perfection of the classification for the training, testing, and validation subsets was 0.8924, 0.7834, and 0.9213, respectively. The network structure (e.g., the weights between various neurons) is displayed in Appendix A. The target vs. output predictions on the validation subset are displayed in Figure 6.

The global sensitivity analysis showed a similar picture as in case of the classification of morphology. The most important factor affecting the SSA was found to be the calcination temperature, followed by the structure of the precursor materials (where also the amorphous content was the most predominant), the reaction temperature, and acid content of the washing liquid, respectively. The SSA of the starting precursor also plays a considerable role in the SSA of the product but, regarding the size of this effect, the different models exhibited some inconsistencies, which may be due to that fact that most of the available papers discussed limited information on the physical properties of the starting materials or the available data showed high versatility. One of the most used precursors was TiO_2_ P25 (Degussa AG or Evonik AG, Essen, Germany), but its reported properties highly varied in the literature. The specification datasheet refers to a unique anatase/rutile ratio, which varied between 80.21 ± 5.59% anatase and 17.40 ± 5.39% rutile contents, respectively, while some publications also reported 4.75 ± 5.72% amorphous content. Similarly, the average particle size of the anatase and rutile components varied between 10–48 nm and 14.4–51 nm, respectively. The SSA of the product was found to be 52.9 ± 12.08 m^2^/g [135,136,137].

According to the discussed findings, a detailed flow chart (Figure 7) is presented as a systematic pathway for TNT transformation into possible drug carriers, indicating the CMAs and CPPs that could affect this procedure and its multiple steps according to the authors’ previous experience and precise screening of the published articles discussing this topic.

## 4. Conclusions

The hydrothermal treatment method holds huge promise for the future development of titanate nanotubes with tailored specifications intended for therapeutic purposes. Therefore, suggesting a systematic approach for transforming hydrothermally synthesized TNTs for pharmaceutical applications should be adopted. This could be achieved through the QbD approach and risk assessment tools, which have recently created a new chapter in the development of the pharmaceutical industry. Moreover, creating clear guidelines for TNT preparation using the QbD approach could open the door for accelerating the scale-up process and transferring their manufacture/use to the next level, especially with the existence of too many factors affecting this complicated procedure. This would have a huge positive impact on the effort needed, time, and cost.

In this study, the most important QTPP elements of hydrothermally synthesized TNTs were identified. Then, the CQAs of nanotubes, CMAs, and CPPs that could have an impact on hydrothermal treatment during titanate nanotube preparation and subsequent steps for their functionalization and loading with APIs were also collected and evaluated.

The CQAs, CMAs, and CPPs with the highest influence and most important impact were identified using Pareto analysis. Surface characteristics, morphology, and SSA were the most significant CQAs. The pH of the washing solution, reaction temperature, and reaction time were the most important CPPs, while precursor concentration, functionalizing agent, and alkaline medium concentration were the highest influencing factors among CMAs.

One of the main targets of this work was to collect and analyze all of the available data into a single paper, thus establishing a solid base that could be of great importance to other researchers as it could be used as a starting point for deeper investigation and further development, and to establish an ANN-based model for the prediction of the most important CQAs. In conclusion, the available literature contains numerous inconsistencies, especially regarding the physical properties of the precursor materials and the parameters of the washing step, which would enable the building of more robust models for the prediction of the expected research outcome. Nevertheless, it was possible to build models for the prediction of product morphology and SSA with acceptable prediction performance.

## Figures and Tables

**Figure 1 pharmaceutics-17-00047-f001:**
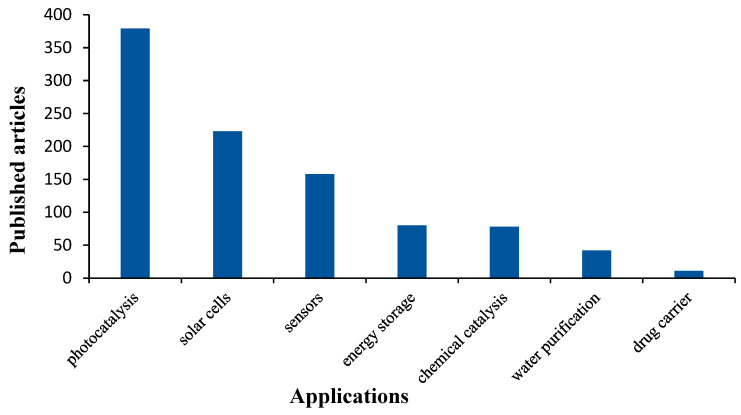
The number of publications regarding TNT usage in different applications (based on the Web of Science database. Available online: https://www.webofscience.com/wos/woscc/basic-search (accessed on 27 April 2023).

**Figure 2 pharmaceutics-17-00047-f002:**
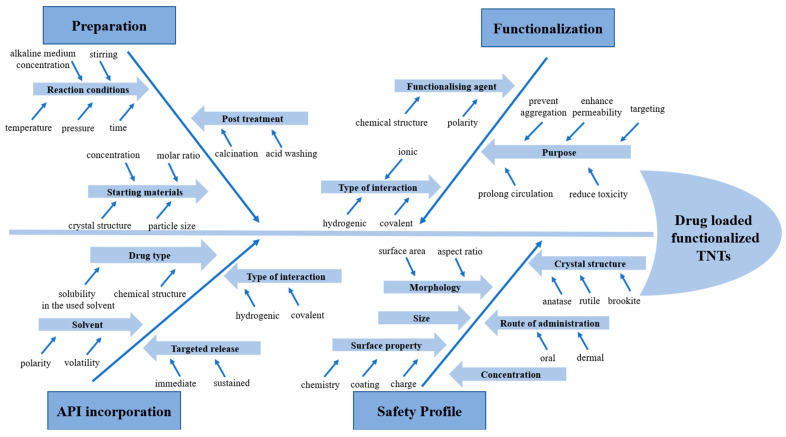
Ishikawa diagram of the factors affecting the procedure of TNT transformation into possible drug carriers. The arrows are representing the structure of the “Fishbone”.

**Figure 3 pharmaceutics-17-00047-f003:**
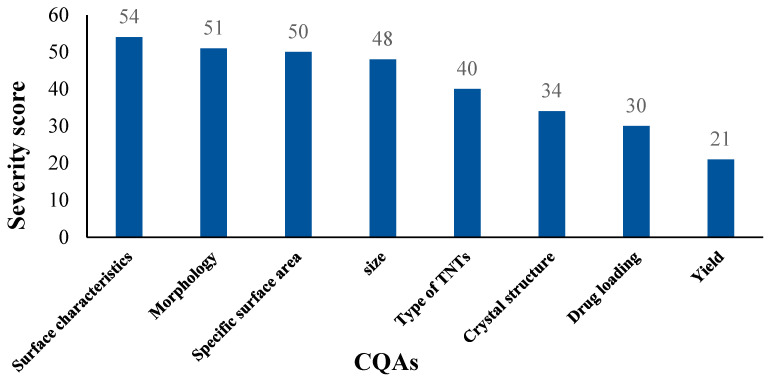
Ranking of CQAs of hydrothermally synthesized TNTs as possible drug carriers.

**Figure 4 pharmaceutics-17-00047-f004:**
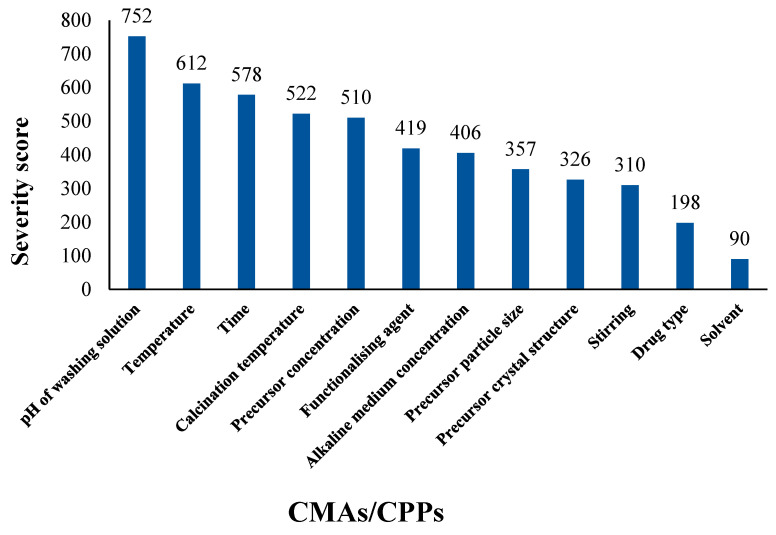
Ranking of CMAs and CPPs during transformation of hydrothermally synthesized TNTs into possible drug carriers.

**Figure 5 pharmaceutics-17-00047-f005:**
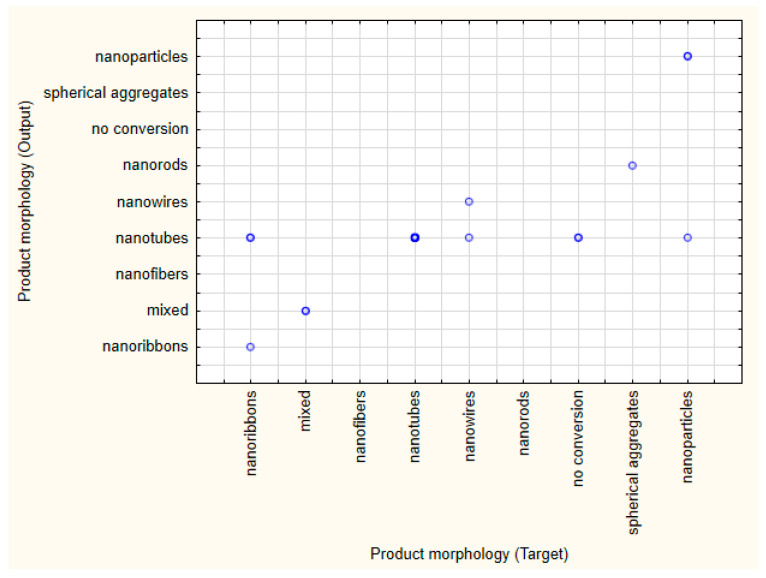
Target vs. output results of the classification of particle morphology on the validation subset. (light and dark blue dots represents one single case and multiple cases, respectively).

**Figure 6 pharmaceutics-17-00047-f006:**
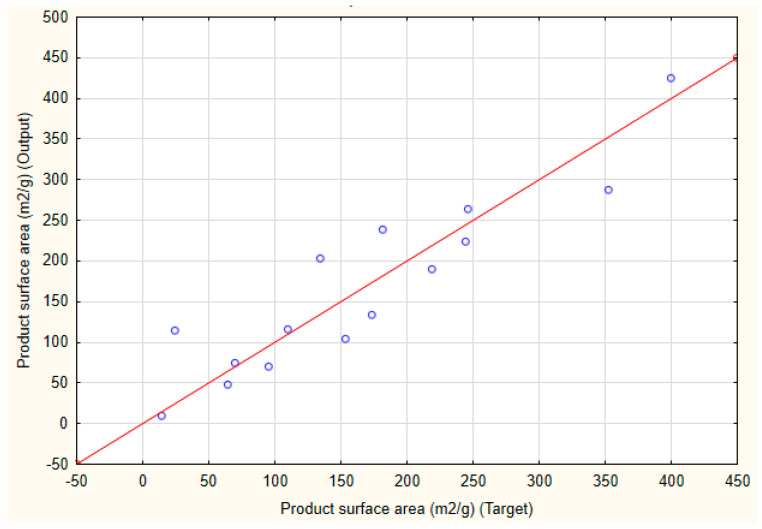
Target vs. output results of the specific surface area on the validation subset. (red line represents the ideal target vs. output relation while blue dots represent the individual predictions).

**Figure 7 pharmaceutics-17-00047-f007:**
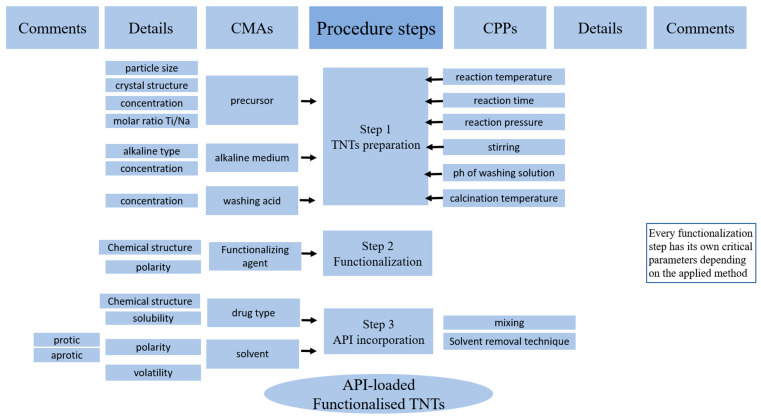
Critical material attributes and critical process parameters that affect TNT transformation into possible drug carriers.

**Table 1 pharmaceutics-17-00047-t001:** QTPPs of titanate nanotubes prepared by the hydrothermal treatment method and intended for pharmaceutical applications.

QTPP	Details	Comments
Physical attributes	The most significant attribute is the tubular morphology	Provides unique characteristics compared to traditional spherical counterparts
Therapeutic effect	Based on the used API	Problems related to some APIs could be solved using TNTs as carriers, like poor tabletting or poor wettability, which would positively affect their therapeutic outcomes
Pharmacokinetics	Absorption, distribution, metabolism, and excretion	Could be optimized/enhanced by surface modification
Safety profile	Accumulation is the highest concern	Affected mainly by physical attributes and needs further investigation
Stability	During processing	TNTs need to be stable and preserve their original characteristics, especially their unique tubular structure and surface properties
Drug release	Depends on requested indication and type of surface modification	Immediate or sustained release could be achieved

**Table 2 pharmaceutics-17-00047-t002:** Critical quality attributes (CQAs) of hydrothermally synthesized titanate nanotubes as possible drug carriers.

CQA	Details	Comments
Type of TNT	Na-TNTs or H-TNTS	Affects several characteristics of TNTs, like morphology, surface area, stability, and safety
Size	Could be regulated by varying preparation conditions	Not large enough so they can be easily detected by the immune system
Surface characteristics	No modification	Aggregation problem, rapid elimination, poor pharmacokinetics profile, higher toxicity
	Functionalized surface	Prolonged circulation, better permeability and PK, less toxicity, targeting is a possibility
Morphology	Tubular	Self-organized or randomly distributed depending on the preparation method
Specific surface area (SSA)	Affect different applications and surface modification	Tubular morphology presents higher surface area compared to traditional nanoparticles
Crystal structure	Affect applications and safety profile	-
Yield	Affects cost efficiency	Critical for scaling up the process

**Table 3 pharmaceutics-17-00047-t003:** Interdependence between CQAs and QTPPs. (Red (H), Yellow (M) and Green (L) are associated with high, medium and low risk, respectively).

	QTPP	Drug Release	Stability	Safety Profile	Therapeutic Effect	Pharmacokinetics	Physical Attributes
CQA							
Size		M	M	H	H	H	H
Crystal structure		L	H	H	L	L	M
Morphology		M	H	H	H	H	H
Specific surface area		H	M	H	H	H	H
Yield		L	L	L	L	L	L
TNT type		L	H	H	L	M	H
Surface characteristics		H	H	H	H	H	M
Drug loading		L	L	L	H	L	M

**Table 4 pharmaceutics-17-00047-t004:** Interdependence between CQAs, CMAs, and CPPs. (Red (H), Yellow (M) and Green (L) are associated with high, medium and low risk, respectively).

**Process**		**Preparation Reaction**							
	**CPP/CMA**	**Temperature**	**Alkaline Medium Concentration**	**Time**	**Stirring**	**Precursor Particle Size**	**Precursor Crystal Structure**	**Precursor Concentration**	
**CQA**									
Size		H	L	H	M	L	L	H	
Crystal structure		H	L	M	L	L	H	L	
Morphology		H	H	H	M	H	H	H	
Specific surface area		H	H	H	L	H	L	H	
Yield		H	H	H	H	L	L	H	
TNT type		L	L	L	L	L	L	L	
Surface characteristics		L	L	L	L	L	L	L	
Drug loading		L	L	L	L	L	L	L	
**Process**		**Post Treatment**				**Surface Modification**		**API Incorporation**	
	**CPP/CMA**	**pH of Washing Solution**		**Calcination Temperature**		**Functionalizing Agent**		**Drug Type**	**Solvent**
**CQA**									
Size		H		L		M		L	L
Crystal structure		M		H		L		L	L
Morphology		H		H		L		L	L
Specific surface area		H		H		L		L	L
Yield		H		H		L		L	L
TNT type		H		L		L		L	L
Surface modification		L		L		H		M	L
Drug loading		L		L		M		H	H

## Data Availability

Data used in risk assessment and ANN modeling are available in public databases, while the exact dataset used for training, testing, and validating the ANNs are displayed in the Appendix A.

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
