# Peer review of "Quality by Design-Based Methodology for Development of Titanate Nanotubes Specified for Pharmaceutical Applications Based on Risk Assessment and Artificial Neural Network Modeling"

_pharmaceutics, 2025, doi:10.3390/pharmaceutics17010047_

Round 1

Reviewer 1 Report

Comments and Suggestions for Authors

The authors presented an intriguing approach to a Quality by Design-based methodology for the development of titanate nanotubes tailored for pharmaceutical applications, utilizing risk assessment and artificial neural network modeling. This study demonstrates a commendable level of originality and provides experimental evidence that aligns with the publication standards of Pharmaceutics. However, a few minor revisions are necessary for final acceptance.

My comments as follows:

1. In line 272, it is recommended to use "fifteen" instead of "15" at the start of the sentence.

3.  Please replace "sodium ions" with "Na+" and verify the notation for magnesium, ensuring it is represented as "Mg+" in lines 371 and 372.

2. In line 465, the authors indicated that the alkaline substance utilized was NaOH; however, clarification is needed regarding the acid present in the washing liquid.

4- “Any word you intend to abbreviate should be spelled out at first occurrence. The first spelled out occurrence should be followed by the abbreviation in parenthesis” When the authors mentioned this word after that, they should write its abbreviation only. These comments should be carefully considered in the whole manuscript.

For examples:

i- The first occurrence of active pharmaceutical ingredient should be spelled out followed by the abbreviation in parentheses (API) (line 120).

ii- Please, delete the words " Quality by Design " (line 132).

iii- Please, delete the words " active pharmaceutical ingredients " (line 226).

iv- Please, delete the words " key material attributes " and “process parameters” (line 387).

v- Please, delete the words “titanate nanotubes" and use TNTs instead (line 388).

vi- The first occurrence of risk estimation matrix should be spelled out followed by the abbreviation in parentheses (REM) (line 396).

vii- Please, delete the dots between the letters (S.S.A.) (line 400)

viii- Please, delete the words “critical quality attributes” and use CQAs instead (line 424).

ix- Please, delete the words “critical material attributes and critical process parameters” and use CQAs and CPPs instead, respectively (line 456). Please, do the same changes (line 522)

Author Response

Response to comments of Reviewer 1:

The authors presented an intriguing approach to a Quality by Design-based methodology for the development of titanate nanotubes tailored for pharmaceutical applications, utilizing risk assessment and artificial neural network modeling. This study demonstrates a commendable level of originality and provides experimental evidence that aligns with the publication standards of Pharmaceutics. However, a few minor revisions are necessary for final acceptance.

We would like to the thank the good evaluation and valuable comments of the Reviewer, that helped us to improve the quality of our manuscript.

My comments as follows:

  1. In line 272, it is recommended to use "fifteen" instead of "15" at the start of the sentence.

The term 15 was replaced as requested.

  1. Please replace "sodium ions" with "Na+" and verify the notation for magnesium, ensuring it is represented as "Mg+" in lines 371 and 372.

The sentence was revised as requested.

  1. In line 465, the authors indicated that the alkaline substance utilized was NaOH; however, clarification is needed regarding the acid present in the washing liquid.

In most cases HCl was used in the washing step, but in some cases HNO3 was applied for washing, as both acids contains only one H+, there was no difference in the activity related to the concentration. This was clarified in the text as requested.

4- “Any word you intend to abbreviate should be spelled out at first occurrence. The first spelled out occurrence should be followed by the abbreviation in parenthesis” When the authors mentioned this word after that, they should write its abbreviation only. These comments should be carefully considered in the whole manuscript.

Thank you for this valuable comment we made a careful readthrough on the whole manuscript.

For examples:

i- The first occurrence of active pharmaceutical ingredient should be spelled out followed by the abbreviation in parentheses (API) (line 120).

Thank you pointing this, we indroduced the term active pharmaceutical ingredient in line 226 but it is right that the first mentioning was made earlier.

ii- Please, delete the words " Quality by Design " (line 132).

It was deleted as requested.

iii- Please, delete the words " active pharmaceutical ingredients " (line 226).

It was deleted as requested.

iv- Please, delete the words " key material attributes " and “process parameters” (line 387).

It was deleted as requested.

v- Please, delete the words “titanate nanotubes" and use TNTs instead (line 388).

It was changed as requested.

vi- The first occurrence of risk estimation matrix should be spelled out followed by the abbreviation in parentheses (REM) (line 396).

It was made as requested.

vii- Please, delete the dots between the letters (S.S.A.) (line 400)

It was made as requested.

viii- Please, delete the words “critical quality attributes” and use CQAs instead (line 424).

It was made as requested.

ix- Please, delete the words “critical material attributes and critical process parameters” and use CQAs and CPPs instead, respectively (line 456). Please, do the same changes (line 522)

It was made as requested.

Reviewer 2 Report

Comments and Suggestions for Authors

This work offers a comprehensive overview of the advancements in the study of TNTs and their applications and future prospects in the pharmaceutical industry.  Although the article is more of a review overall, it gives its own insights on some important parameters involved in TNTs preparation.  It employs a machine learning approach to analyze critical parameters in the preparation of TNTs, providing scientifically validated recommendations that can greatly benefit the reader.  It is recommended that the journal accept the paper after minor revision.

1.  critical material attributes (CMAs), critical process parameters (CPPs), The key material attributes (CMAs) and process parameters (CPPs), please unify the full description of the above abbreviations

2.  The contents of Table 2 are not complete, please supplement

3.  The author is suggested to add relevant content of modeling and result verification and evaluation

4.  I recommend that authors consider sharing their models or code.

Author Response

Response to comments of Reviewer 2:

This work offers a comprehensive overview of the advancements in the study of TNTs and their applications and future prospects in the pharmaceutical industry.  Although the article is more of a review overall, it gives its own insights on some important parameters involved in TNTs preparation.  It employs a machine learning approach to analyze critical parameters in the preparation of TNTs, providing scientifically validated recommendations that can greatly benefit the reader.  It is recommended that the journal accept the paper after minor revision.

We would like to the thank the good evaluation and valuable comments of the Reviewer, that helped us to improve the quality of our manuscript.

  1.  critical material attributes (CMAs), critical process parameters (CPPs), The key material attributes (CMAs) and process parameters (CPPs), please unify the full description of the above abbreviations

The unification was made as requested.

  1.  The contents of Table 2 are not complete, please supplement

The alignment of Table 2 was corrected.

  1.  The author is suggested to add relevant content of modeling and result verification and evaluation

According to our opinion the relevant data of the modelling (perfection e.g. the goodness of fit of the observed vs. predicted data of the various susets) and the result verification (graphical represenention of the predicition accuracy of the external validation subset) was appropriately displayed in the text, according to the methods which is usual in similar articles.

  1.  I recommend that authors consider sharing their models or code.

As we used the commercially available Statistica software the code of the system cannot be shared, but the supplementary material was completed with the exact network structure e.g the weights between the neurons of the best networks as it is indicated in the text of the revised manuscript. (see lines 500 and 523)

Reviewer 3 Report

Comments and Suggestions for Authors

The manuscript by Saker R. et al. entitled: "Quality by Design-based Methodology for Development of Titanate Nanotubes Specified for Pharmaceutical Applications  based on Risk Assessment and Artificial Neural Network Modelling" is an interesting and valuable study on the computation of factors influencing the production of pharmaceutical titanate nanotubes using the quality-by-design approach.

The manuscript is well written and well structured; it uses clear language, and the results are well presented.

The authors follow a structure that flows easily, is logical and is presented in a clear manner.

However, the following claim in the Introduction section (lines 40-43: Their unique tubular morphology and higher surface area compared to their spherical precursor in addition to the existence of hydroxyl groups on their vast surface provide them with preferred features for medical/pharmaceutical applications, such as good wettability and biocompatibility) is not entirely correct, as their good biocompatibility is not due to their unique tubular morphology and higher surface area.

The authors state their experience in the field of titanate nanotubes in too many sentences in the introduction section and even cite many of their previous articles just to reinforce this, which I find inappropriate. Honestly, every time a research team plans a new study, they either rely on their experience or on the data available in the literature, so it is quite normal to have experience when developing an article. Also, all researchers collect a lot of data from the literature when they start a study or a manuscript.

This manuscript is a mixture of a literature review and an original article, but it is very interesting, and the combination is useful.

The tables and figures are very well developed and suggestive of the chosen topic, but Table 2 is not adequately fitted into the page.

Author Response

Response to comments of Reviewer 3:

The manuscript by Saker R. et al. entitled: "Quality by Design-based Methodology for Development of Titanate Nanotubes Specified for Pharmaceutical Applications  based on Risk Assessment and Artificial Neural Network Modelling" is an interesting and valuable study on the computation of factors influencing the production of pharmaceutical titanate nanotubes using the quality-by-design approach.

The manuscript is well written and well structured; it uses clear language, and the results are well presented.

The authors follow a structure that flows easily, is logical and is presented in a clear manner.

We would like to the thank the good evaluation and valuable comments of the Reviewer, that helped us to improve the quality of our manuscript.

However, the following claim in the Introduction section (lines 40-43: Their unique tubular morphology and higher surface area compared to their spherical precursor in addition to the existence of hydroxyl groups on their vast surface provide them with preferred features for medical/pharmaceutical applications, such as good wettability and biocompatibility) is not entirely correct, as their good biocompatibility is not due to their unique tubular morphology and higher surface area.

We agree with the Reviewer’s opinion. and the referred paragraph was revised.

The authors state their experience in the field of titanate nanotubes in too many sentences in the introduction section and even cite many of their previous articles just to reinforce this, which I find inappropriate. Honestly, every time a research team plans a new study, they either rely on their experience or on the data available in the literature, so it is quite normal to have experience when developing an article. Also, all researchers collect a lot of data from the literature when they start a study or a manuscript.

We would like to debate with the Reviewers opinion about the extensive self citations, as less than 10% of the cited article refers to our own work, and since the number of available literature about the pharmaceutical use of TNTs is highly limited, the appropriate representation of the sate of the art would be challenging without the mentioning of these articles.

This manuscript is a mixture of a literature review and an original article, but it is very interesting, and the combination is useful.

The tables and figures are very well developed and suggestive of the chosen topic, but Table 2 is not adequately fitted into the page.

The alignment of Table 2 was corrected.